# Composition of time in movement behaviors and weight change in Latinx, Black and white participants

**Erika Rees-Punia** **\*, Mark A. Guinter, Susan M. Gapstur, Ying Wang, Alpa V. Patel**

American Cancer Society, Atlanta, Georgia, United States of America

\* Erika.Rees-Punia@cancer.org

## Abstract

### Background

The relationship between time-use behaviors and prospective weight change is poorly understood.

### Methods

A subset of Cancer Prevention Study-3 participants (n = 549, 58% women, 66% non-Latinx white) self-reported weight in 2015 and 2018 and completed an accelerometer protocol for seven days. Sedentary time, sleep, light, moderate, and vigorous intensity physical activity (PA) were treated as a compositional variable and multiple linear regression was used to examine associations between activity composition and weight change stratified by sex and race/ethnicity. Compositional isotemporal substitution analysis was used to quantify change in weight associated with reallocating 30 min./day.

### Results

Activity composition was associated with weight change among women ($p = 0.007$), but not men ($p = 0.356$), and among Latinx ($p = 0.032$) and white participants ($p = 0.001$), but not Black participants ($p = 0.903$). Replacement of 30 min./day sedentary time with moderate-vigorous PA was associated with 3.49 lbs. loss (-6.76, -0.22) in Latinx participants and replacement with sleep was associated with 1.52 (0.25, 2.79) and 1.31 (0.40, 2.21) lbs. gain in white women and men.

### Conclusion

The distribution of time spent in daily behaviors was associated with three-year weight change in women, Latinx, and white participants. This was the first longitudinal compositional study of weight change; thus, more studies are needed.

**Data Availability Statement:** The authors confirm that some access restrictions apply to the data underlying the findings. Data are from the Cancer Prevention Study 3 and are available from the

American Cancer Society by following the ACS Data Access Procedures (https://www.cancer.org/content/dam/cancer-org/research/epidemiology/cancer-prevention-study-data-access-policies.pdf) for researchers who meet the criteria for access to confidential data.

**Funding:** The American Cancer Society (ACS) funds the creation, maintenance, and updating of the Cancer Prevention Study-3. The authors confirm that some access restrictions apply to the data underlying thefindings. Data are from the Cancer Prevention Study 3 and are available from the American Cancer Society by following the ACS Data Access Procedures (https://www.cancer.org/content/dam/cancer-org/research/epidemiology/cancer-prevention-study-data-access-policies.pdf) for researchers who meet the criteria for access to confidential data. Requests may be sent to Alpa V. Patel, PhD.

**Competing interests:** The authors have declared that no competing interests exist.

## Introduction

Research supports that physical activity [1], sedentary behavior [2, 3], and sleep [4] may be "independently" associated with the development of overweight or obesity. However, given that these time-use behaviors comprise the entire 24-hour day, it is more likely that physical activity, sedentary time, and sleep are associated with weight change in a unique, co-dependent manner [5]. Traditional statistical analyses including one single time-use behavior in isolation, or all time-use behaviors together, ignore assumptions regarding the perfect collinearity of the behaviors [5, 6]. To better understand the relationships between the distribution of time-use behaviors and weight without ignoring their compositional and constrained nature, the associations between physical activity, sedentary time, sleep, and weight change should be studied relative to each other, rather than in isolation or with partial adjustment for the other behaviors.

Compositional data analysis methods were developed to analyze constrained data that make up parts of a finite whole (e.g., a 24-hour day) [7]. Compositional techniques have long been implemented in nutritional epidemiology, pharmacology, and other areas of health science [8–10], but have only recently been used in physical activity epidemiology. Among the few prior studies using compositional analyses to explore associations with weight status or body composition, all are cross-sectional [11–15], and many include only children [11, 13–15]. This has left gaps in our understanding of the associations between healthful time-use and longitudinal weight change, especially among adult men and women (who may experience differing weight change responses to physical activity in a real-world setting) of various racial/ethnic backgrounds [16].

The primary purpose of this study was to use compositional data analysis to investigate the prospective relationships between accelerometer-assessed time-use behaviors and three-year change in self-reported weight in male and female Latinx, Black, and white adults in a substudy of the Cancer Prevention Study-3. Using compositional isotemporal substitution analyses, associations between reallocations of time spent sedentary and three-year weight change were also explored.

## Methods

### Study population

The Cancer Prevention Study-3 (CPS-3) is a U.S.-based prospective cohort study of cancer incidence and mortality initiated by the American Cancer Society (ACS). All participants provided written informed consent. All aspects of the Cancer Prevention Study-3 are approved by the Emory University Institutional Review Board. Approximately 304,000 participants aged 30 to 65 years with no history of cancer were enrolled at ACS fundraising events or community enrollment drives between 2006 and 2013 [17]. In 2015, CPS-3 participants were stratified by sex and race/ethnicity and randomly invited to participate in the CPS-3 Activity Validation Sub-study (AVSS). Among the 10,000 participants invited to the AVSS, 1,801 participants pre-registered and consented to participate. The original intent of the AVSS was to validate the physical activity and sedentary time questionnaire in multiple racial/ethnic groups, therefore the first 300 non-Latina white female, 150 non-Latino white male, 150 Latinx, and 150 non-Latinx African-American/Black participants to complete the routine 2015 CPS-3 follow-up survey were enrolled into the AVSS (AVSS discussed in more detail elsewhere) [18, 19]. Participants were excluded from the current analysis if they were missing self-reported weight at either time point (missing 2015, n = 12; missing 2018, n = 171), or did not have at least three days of valid accelerometer (n = 12) or sleep diary (n = 6) data. The remaining 549 participants

were included in this study. Excluded participants did not significantly differ from included participants in age, BMI, sex, or race/ethnicity (S1 Table).

## Exposure measures

Data collection for the CPS-3 AVSS occurred over a one-year period (2015), which was split into four quarters. During two non-consecutive quarters (Q1/Q3 or Q2/Q4), participants wore an Actigraph GT3x$^+$ accelerometer on the hip aligning with the midline of the non-dominant thigh. Participants were instructed to wear the device for seven consecutive days during all waking hours, except when bathing or swimming, during both assigned quarters (i.e., seven days in Q1/seven days in Q3 OR seven days in Q2/seven days in Q4). Data were collected at a sampling rate of 30 Hz and were downloaded and stored in one second epochs (required for Choi and Sojourn algorithms) for subsequent analyses.

Actigraph data were processed using the Choi algorithm to calculate accelerometer wear time [20, 21]. Days with less than 14 hours of wear time per day were not considered valid and excluded from analyses, as they were unlikely to represent a full waking period. The sojourn-3 axis algorithm, which is a hybrid machine-learning, neural network, and decision tree analysis algorithm, was used to estimate daily sedentary, light intensity physical activity (LPA), moderate intensity physical activity (MPA), and vigorous intensity physical activity (VPA) time [22].

Participants were instructed to remove the accelerometers before bed, thus, time spent sleeping was estimated from 24-hour diaries. Participants were asked to code their daily time-use behaviors, including sleep, in 15-minute epochs for the seven days concurrent with accelerometer wear. Average time spent sleeping was calculated from the diaries for participants with at least three valid days of diary data concurrent with accelerometer wear. Diary data was not otherwise used for sedentary time, LPA, MPA, or VPA, as accelerometer-assessed data were available.

## Outcome measures

At the start of the CPS-3 AVSS (2015), participants received a four-page pre-study survey that included self-reported current weight (self-weighed and self-reported in pounds [lbs.]). As a part of the routine CPS-3 follow-up, participants once again self-reported weight on their 2018 tri-annual survey. Three-year weight change was calculated as the difference between 2018 and 2015 weight. Weight change was assessed continuously for regression models, and weight change groups were used for descriptive analyses: weight loss (>-2 lbs.), weight stable ($\leq \pm2$ lbs.), and weight gain (>2 lbs.).

## Statistical analysis

To address the potential role of all movement behaviors (especially VPA) in weight loss [23, 24], a 5-part composition was considered for the main analysis; this was possible as the average daily VPA was at least 0.5 min/day for all included participants (range: 0.51–91.7 min/day). Accelerometer-measured sedentary time, LPA, MPA, and VPA and self-reported sleep time were treated as co-dependent, compositional variables for all analyses. These time-use behaviors were first transformed to represent ratios of parts using an isometric log-ratio (*ilr*) coordinate system [7]. After *ilr* transformations were applied, the data could be analyzed using any standard statistical technique that is valid under the assumptions applying to data in real space, as *ilr* transformations preserve all metric properties of data in coordinates with a non-singular covariance matrix [6]. Although some participants did not have exactly 1440 total minutes of data per day (mean = 1430 min/day, Q1 = 1391 min/day, Q3 = 1446 min/day), compositional data are scale invariant, thus, the relative structure (i.e., the ratios between

behaviors) remains the same even if the observed subset of behaviors is not closed to a specific time [6].

The geometric mean of time spent in each time-use behavior and the log-ratio variances between each pair of behaviors was calculated. For the main analyses, separate multiple linear regression models were first used to examine associations between the 5-part time-use composition (expressed as isometric log-ratios) and three-year weight change (absolute change and percent change modeled continuously) stratified by sex or race/ethnicity and, where sample size allowed, by both sex and race/ethnicity. Models were checked to ensure the assumptions of linearity, normality, homoskedasticity and leverage were not violated. A Wald chi square ANOVA type II test of the multiple linear regression models was used to assess the significance of the 5-part time use composition.

Compositional isotemporal substitution analysis was then applied to quantify the change in weight associated with the reallocation of 30 minutes of sedentary time to an equivalent amount of time in another behavior, while keeping time spent in the remaining behaviors constant [25]. Compositional isotemporal substitution analyses explored the replacement of sedentary time with sleep, LPA, or moderate-vigorous physical activity combined (as average time in VPA alone was well under 30 min/day (mean: 13 min/day) and would therefore result in negative time when modeling 30 minutes of replacement).

All models were sex and/or race/ethnicity stratified, and the following covariates from 2015 CPS-3 surveys were identified *a priori*: age, race/ethnicity (White, Black, Latinx; for sex stratified models), sex (for race/ethnicity stratified models), average daily caloric intake (continuous; estimated from the food frequency questionnaire), comorbidity score (sum of comorbidities including hypertension, hypercholesterolemia, and type 2 diabetes), and height (continuous). Smoking status information was available, however very few participants smoked (2% current smokers), so smoking status was not included in the models. A sensitivity analysis restricting to participants who were overweight or obese at baseline was also conducted, as normal weight participants may be less likely to experience a change in weight over a three-year period. All analyses were conducted in Rstudio v. 3.5.2 using the Compositions package, and the significance level was set at $P < 0.05$ [26].

## Results

Overall, participants spent 41.6% of the day sedentary (599 min/day), 18.4% in LPA (264 min/day), 4.0% in MPA (57 min/day), 0.6% in VPA (9 min/day), and 35.5% sleeping (511 min/day) on average. Women spent less time sedentary and more time in LPA compared to men (S2 Table). Black participants spent more time sedentary than Latinx and white participants, and Latinx participants spent more time in LPA than Black and white participants (S3 Table).

Over the three-year period, 47.2% of women and 47.6% of men gained weight (Table 1). Similarly, 46.9% of Latinx, 53.4% of Black participants, and 45.8% of white participants gained weight over three years (Table 2). Women who lost weight over the three-year period were more likely to have at least one comorbidity (49%) compared to women who maintained (34%) or gained weight (35%). On the other hand, Black participants who lost (71%) or maintained weight (69%) were more likely to have at least one comorbidity compared to those who gained weight (49%). Similar patterns among Latinx and white participants who lost weight were not observed. Demographic characteristics were otherwise similar across weight change groups.

The activity composition (sedentary time, LPA, MPA, VPA, and sleep) was significantly associated with three-year weight change among women ($p = 0.007$), but not men ($p = 0.356$), and Latinx ($p = 0.032$) and white participants ($p = 0.001$), but not Black participants ($p = 0.903$;

**Table 1. Baseline characteristics by sex.**

| Characteristics | Women (n = 318) | | | Men (n = 231) | | |
|---|---|---|---|---|---|---|
| | Loss* (n = 111) | Maintain (n = 57) | Gain** (n = 150) | Loss (n = 84) | Maintain (n = 37) | Gain (n = 110) |
| | Arithmetic Mean (SD) | | | | | |
| Age (years) | 58 (9) | 56 (10) | 54 (10) | 59 (10) | 58 (12) | 56 (10) |
| Baseline weight (lbs.) | 167 (35) | 150 (37) | 163 (39) | 200 (35) | 184 (38) | 192 (30) |
| 3-year weight change (lbs.) | -9 (12) | 0.1 (0.7) | 11 (12) | -8 (6) | 0.1 (0.6) | 10 (8) |
| Energy intake (kcals) | 1965 (650) | 1914 (741) | 1999 (596) | 2030 (666) | 1856 (631) | 1904 (718) |
| | N (%) | | | | | |
| Race/Ethnicity | | | | | | |
| White/Non-Latinx | 71 (64%) | 41 (72%) | 99 (66%) | 58 (69%) | 28 (76%) | 68 (62%) |
| Black/Non-Latinx | 21 (19%) | 9 (16%) | 32 (21%) | 14 (17%) | 4 (11%) | 23 (21%) |
| Latinx | 19 (17%) | 7 (12%) | 19 (13%) | 12 (14%) | 5 (13%) | 19 (17%) |
| Current smoker | 2 (2%) | 2 (4%) | 1 (1%) | 3 (4%) | 1 (3%) | 2 (2%) |
| Baseline body mass index | | | | | | |
| Underweight | 0 (0%) | 3 (5%) | 1 (1%) | 0 (0%) | 1 (3%) | 0 (0%) |
| Normal | 37 (33%) | 35 (61%) | 68 (45%) | 24 (29%) | 15 (41%) | 34 (31%) |
| Overweight | 41 (37%) | 13 (23%) | 44 (29%) | 34 (41%) | 16 (43%) | 51 (46%) |
| Obese | 33 (30%) | 6 (11%) | 37 (25%) | 26 (31%) | 5 (14%) | 25 (23%) |
| Comorbidity score[†] | | | | | | |
| 0 | 57 (51%) | 38 (67%) | 97 (65%) | 44 (52%) | 14 (38%) | 48 (44%) |
| 1 | 30 (27%) | 12 (21%) | 31 (21%) | 21 (25%) | 13 (35%) | 41 (37%) |
| 2+ | 24 (22%) | 7 (12%) | 22 (15%) | 19 (23%) | 10 (27%) | 21 (19%) |

*Lost at least two lbs.

**Gained at least two lbs.

[†]Sum of comorbidities including hypertension, hypercholesterolemia, and diabetes.

Table 3). The associations between the activity composition and percent weight change were very similar (S4 Table). There was significant interaction by race/ethnicity and sex ($p = 0.03$), though sample sizes were insufficient for joint analyses by sex and race/ethnicity among Latinx and Black participants. Stratified analyses for white women (n = 211) and white men (n = 154) revealed that the activity composition was associated with three-year weight change among white women ($p = 0.003$) and among white men ($p = 0.036$). Results restricting to participants who were overweight or obese at baseline were largely similar (S5 Table).

Relative time spent in each movement behavior for participants in each weight change category is presented for four sex and racial/ethnic groups where composition of time was statistically significantly associated with weight change in Fig 1. These data are presented as the log-ratio between each weight change groups' compositional mean and the overall compositional mean, such that positive and negative bars reflect relative mean values above and below the overall mean. Time spent sedentary, in LPA, MPA, and sleeping were relatively similar across all three weight change categories for all examined sex and racial/ethnic groups. However, women who maintained their weight spent a larger proportion of time in VPA compared to those who gained weight, and those who lost weight spent less time in VPA. Particularly among Latinx, those who maintained their weight spent a larger proportion of time in VPA.

Compositional isotemporal substitution analyses were carried out for all women, Latinx, white women, and white men, as the activity composition was significantly associated with weight change in these four sub-groups at α < 0.05. The estimated differences in weight (pounds) for the reallocation of 30 minutes of sedentary time with equivalent amounts of LPA,

**Table 2. Baseline characteristics by race/ethnicity.**

| | Latinx (n = 81) | | | Black (n = 103) | | | White (n = 365) | | |
|---|---|---|---|---|---|---|---|---|---|
| | Loss*<br>(n = 31) | Maintain<br>(n = 12) | Gain**<br>(n = 38) | Loss<br>(n = 35) | Maintain<br>(n = 13) | Gain<br>(n = 55) | Loss<br>(n = 129) | Maintain<br>(n = 69) | Gain<br>(n = 167) |
| | Average (SD) | | | | | | | | |
| Age (years) | 56 (11) | 54 (6) | 54 (10) | 61 (9) | 56 (14) | 52 (10) | 58 (9) | 57 (11) | 56 (10) |
| Baseline weight (lbs.) | 173 (39) | 149 (22) | 165 (29) | 196 (44) | 175 (27) | 190 (45) | 179 (37) | 163 (45) | 173 (37) |
| 3-year weight change (lbs) | -8 (4) | 0.3 (0.7) | 10 (7) | -8 (8) | 0.0 (0.8) | 11 (9) | -9 (11) | 0.1 (0.7) | 11 (12) |
| Energy intake (kcals) | 2032 (706) | 1886 (478) | 1925 (792) | 1990 (679) | 1769 (700) | 1916 (601) | 1985 (642) | 1915 (733) | 1980 (634) |
| | N (%) | | | | | | | | |
| Women | 19 (61%) | 7 (58%) | 19 (50%) | 21 (60%) | 9 (69%) | 32 (58%) | 71 (55%) | 41 (59%) | 99 (59%) |
| Current smoker | 1 (3%) | 0 (0%) | 0 (0%) | 1 (3%) | 0 (0%) | 2 (4%) | 3 (2%) | 3 (4%) | 1 (1%) |
| Baseline body mass index | | | | | | | | | |
| Underweight | 0 (0%) | 1 (8%) | 0 (0%) | 0 (0%) | 1 (8%) | 0 (0%) | 0 (0%) | 2 (2%) | 1 (1%) |
| Normal | 10 (32%) | 6 (50%) | 15 (39%) | 6 (17%) | 1 (8%) | 16 (29%) | 45 (35%) | 43 (64%) | 71 (42%) |
| Overweight | 13 (42%) | 5 (42%) | 17 (45%) | 9 (26%) | 9 (69%) | 16 (29%) | 53 (41%) | 15 (22%) | 62 (37%) |
| Obese | 8 (26%) | 0 (0%) | 6 (16%) | 20 (57%) | 2 (15%) | 23 (42%) | 31 (24%) | 9 (13%) | 33 (20%) |
| Comorbidity score† | | | | | | | | | |
| 0 | 17 (55%) | 8 (67%) | 20 (53%) | 10 (29%) | 4 (31%) | 28 (51%) | 74 (57%) | 40 (58%) | 97 (58%) |
| 1 | 6 (19%) | 4 (33%) | 11 (29%) | 11 (31%) | 4 (31%) | 18 (33%) | 34 (26%) | 17 (25%) | 43 (26%) |
| 2+ | 8 (26%) | 0 (0%) | 7 (18%) | 14 (40%) | 5 (39%) | 9 (16%) | 21 (16%) | 12 (17%) | 27 (16%) |

*Lost at least two lbs.

**Gained at least two lbs.

†Sum of comorbidities including hypertension, hypercholesterolemia, and diabetes.

MVPA, and sleep are presented in Table 4. The replacement of 30 minutes of sedentary time with any other movement behavior was not associated with a statistically significant difference in weight among all women, however the replacement of sedentary time with sleep was associated with a 1.52 lbs. gain (95% Confidence Interval [CI] = 0.25, 2.79) in white women and a

**Table 3. The association between activity composition (expressed as isometric log ratios) and weight change by sex and race/ethnicity*.**

| Sex | Sum sq. | F | p |
|---|---|---|---|
| Women | **2741** | **3.570** | **0.007** |
| Men | 490 | 1.102 | 0.356 |
| Race/Ethnicity | | | |
| Latinx | **984** | **2.810** | **0.032** |
| Black | 134.5 | 0.259 | 0.903 |
| White | **3392** | **4.748** | **0.001** |
| Joint** | | | |
| White women | **3634** | **4.087** | **0.003** |
| White men | **1195** | **2.649** | **0.036** |

*Results from Wald chi square type II test of linear models. All models are adjusted for age, race/ethnicity or sex, average kcal/day (FFQ estimate), comorbidity score, and height. Results in bold are significant at 0.05.

**Insufficient sample size for joint analyses among Latinx or Black participants.

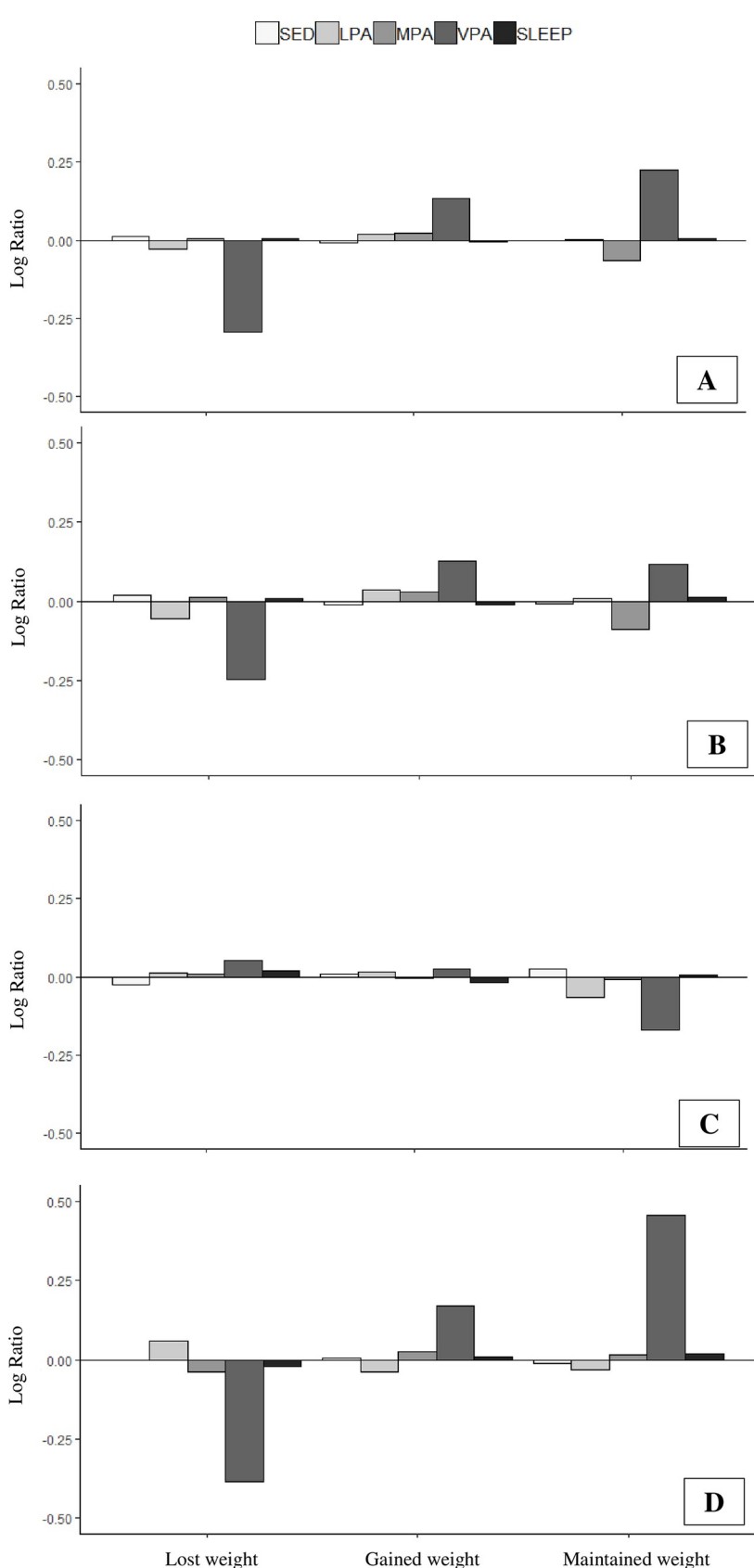

**Fig 1.** Compositional analysis of relative importance of 3-year weight change group mean time spent in each movement behavior in a) all women, b) white women, c) white men, d) Latinx.

1.31 lbs. gain (95% CI = 0.40, 2.21) in white men, and replacement with MVPA was associated with a 3.49 lbs. loss (95% CI = -6.76, -0.22) in Latinx participants.

## Discussion

Overall, the full time-use composition, which includes the distribution of time spent sedentary, in LPA, MPA, VPA, and sleep, was associated with three-year weight change in women overall, Latinx participants, and white women. Secondary analyses further suggest that time in VPA may be important, relative to other behaviors, for weight maintenance. Analysis with the compositional isotemporal substitution framework led to the finding that replacing 30 minutes/day of sedentary time with MVPA was associated with modest weight loss in Latinx participants and replacing 30 minutes/day of sedentary time with sleep was associated with modest weight gain in white men and women.

Other compositional analyses, although cross-sectional in nature, have also reported significant associations between the activity composition and weight in adults [12, 27]. There are also several cross-sectional compositional studies of the relationship between movement behaviors and weight or adiposity in children and adolescents [11, 13–15]. One important difference between the current study and the existing literature, aside from the cross-sectional versus longitudinal designs, is that prior studies to date have not been adequately powered to assess the associations by race/ethnicity. Although studies of the composition of time in movement behaviors are relatively sparse, the 24-hour movement framework is more widely recognized, as is evident in the national physical activity guidelines for Canada and Australia [28, 29].

One striking finding in this study was the relatively low levels of VPA in participants who lost weight, particularly women who lost weight, relative to participants who gained or maintained their weight. One potential explanation for this unexpected finding may be that some of the weight loss was unintentional. This is plausible as women who lost weight in this study were also more likely to have multiple comorbidities at baseline and therefore may have been sick and less active during the three-year study. Alternatively, diet may have played a large role in participant weight loss. All models were adjusted for energy intake estimated from a food frequency questionnaire (FFQ), but some studies suggest that FFQs may not accurately estimate total energy intake [30]. We were also unable to account for diet change during this period. Some studies suggest that while physical activity may not independently contribute to significant weight loss without accompanying dietary changes, it may play a more substantive role in weight maintenance [1]. Thus, it is possible that there may be a role of time-use behaviors in weight maintenance, however dietary changes are needed to elicit significant weight loss. Finally, measurement error in self-reporting weight may also explain these findings, though a recent validation study comparing self-reported weight to measured weight among 2,643 CPS-3 participants found that reported weight was highly correlated with measured weight among men (Pearson $r$ = 0.98) and women (Pearson $r$ = 0.99) [31]. Pertinent to the current study, the validation study also found that mean differences in self-reported compared to measured weight did not differ by race/ethnicity among men (p for difference = 0.528) or women (p = 0.122).

It is clear that changing time spent in one behavior will inevitably influence the time spent in other behaviors throughout the day, meaning that a focus solely on MVPA (thus ignoring other behaviors) may limit our understanding of how physical activity may impact health [6]. The benefits of MVPA, for example, can be attenuated by excessive sedentary time as well as

**Table 4. Estimated difference in 3-year weight change associated with reallocation of 30 minutes.**

| 30 min sedentary time per day reallocated to: | Estimated difference (95% confidence interval), lbs. |
|---|---|
| All women | |
| 30 min LPA | -0.20 lbs. (-0.78, 0.38) |
| 30 min MVPA | -0.71 lbs. (-2.74, 1.31) |
| 30 min sleep | 0.82 lbs. (-0.06, 1.69) |
| White women | |
| 30 min LPA | -0.80 lbs. (-1.61, 0.02) |
| 30 min MVPA | -0.39 lbs. (-2.90, 2.11) |
| 30 min sleep | **1.52 lbs. (0.25, 2.79)** |
| White men | |
| 30 min LPA | -0.20 lbs. (-0.93, 0.53) |
| 30 min MVPA | 0.62 lbs. (-1.64, 2.89) |
| 30 min sleep | **1.31 lbs. (0.40, 2.21)** |
| Latinx | |
| 30 min LPA | 0.28 lbs. (-0.42, 0.99) |
| 30 min MVPA | **-3.49 lbs. (-6.76, -0.22)** |
| 30 min sleep | -1.14 lbs. (-2.29, 0.02) |
| **30 min sleep time per day reallocated to:** | Estimated difference (95% confidence interval), lbs. |
| All women | |
| 30 min LPA | -0.98 lbs. (-1.99, 0.03) |
| 30 min MVPA | -1.50 lbs. (-3.54, 0.04) |
| 30 min sedentary | -0.79 lbs. (-1.66, 0.08) |
| White women | |
| 30 min LPA | **-2.26 lbs. (-3.73, -0.78)** |
| 30 min MVPA | **-1.85 lbs. (-4.42, -0.72)** |
| 30 min sedentary | -1.48 lbs. (-2.73, 0.23) |
| White men | |
| 30 min LPA | **-1.46 lbs. (-2.64, -0.27)** |
| 30 min MVPA | -0.63 lbs. (-2.94, 0.67) |
| 30 min sedentary | -1.28 lbs. (-2.16, 0.39) |
| Latinx | |
| 30 min LPA | 1.38 lbs. (-0.09, 2.67) |
| 30 min MVPA | **-2.40 lbs. (-5.60, -0.08)** |
| 30 min sedentary | 1.12 lbs. (-0.02, 2.26) |
| **30 min LPA per day reallocated to:** | Estimated difference (95% confidence interval), lbs. |
| All women | |
| 30 min sleep | 0.97 lbs. (-0.01, 1.96) |
| 30 min MVPA | -0.56 lbs. (-2.68, 1.56) |
| 30 min sedentary | 0.14 lbs. (-0.39, 0.69) |
| White women | |
| 30 min sleep | **2.20 lbs. (0.76, 3.64)** |
| 30 min MVPA | -0.28 lbs. (-2.38, 2.94) |
| 30 min sedentary | 0.66 lbs. (-0.10, 1.41) |
| White men | |
| 30 min sleep | **1.43 lbs. (0.29, 2.59)** |
| 30 min MVPA | 0.75 lbs. (-1.80, 3.30) |
| 30 min sedentary | 0.10 lbs. (-0.55, 0.76) |
| Latinx | |

(*Continued*)

**Table 4.** (Continued)

| 30 min sleep | **-1.34 lbs. (-2.61, -0.08)** |
|---|---|
| 30 min MVPA | **-3.70 lbs. (-6.76, -0.22)** |
| 30 min sedentary | 0.18 lbs. (-0.84, 0.48) |

Composition of time not associated with men or Black participants, therefore all women, white women, white men, and Latinx participants are included in compositional isotemporal substitution models. All models are adjusted for age, race/ethnicity or sex, average kcal/day (FFQ estimate), comorbidity score, and height.

too much or too little sleep. These joint associations of time-use behaviors suggest that the 24-hour composition, consisting of all time-use behaviors together, needs to be considered. However, it is important to note that compositional analyses are not entirely comprehensive, as they do not allow for consideration of time-of-day, which a limited number of studies suggest may be important for weight change [32], or bouts and clustered patterns of activities.

Strengths of this study include accelerometer-assessed physical activity and sedentary time, with exceptional participant compliance to the accelerometer wear protocol. Additionally, the participant population consisted of 34% racial/ethnic minorities, allowing for stratified analyses of under-represented sub-populations. This study is also strengthened by the consideration of the full 24-hour day and all time-use behaviors at once. To the best of our knowledge, this is the first compositional study of time-use behaviors with prospectively collected weight change.

Limitations of this study include the use of self-reported weight. Further, without information on body composition, we were unable to determine if weight change was due to changes in lean or fat mass. We were also unable to determine intentionality of weight loss, limiting interpretability of the weight loss findings. Additionally, this study focused on only one potential benefit of a healthful balance of time in movement behaviors; it is important to note that movement behaviors likely influence health in many ways beyond weight maintenance or change. As previously noted, another limitation was the limited dietary data. Finally, although the compositional isotemporal substitution models help to better understand which parts of the time-use composition were associated with weight change, it is important to note that these results are modeled and therefore do not describe associations of actual changes in behavior.

## Conclusions

The distribution of time spent in all daily behaviors, including sedentary time, LPA, MPA, VPA, and sleep, was associated with three-year weight change in women overall, Latinx, and white women. Our results suggest that time spent in VPA may be of particular importance for weight maintenance. Additionally, the replacement of 30 minutes of sedentary time with MVPA was associated with modest weight loss in Latinx and replacing 30 minutes/day of sedentary time with sleep was associated with modest weight gain in white men and women. However, more longitudinal studies using compositional analyses are needed to understand the complex relationships between time-use behaviors and weight change.

## Supporting information

**S1 Checklist. STROBE statement—checklist of items that should be included in reports of *cohort studies*.**
(DOCX)

**S1 Table. Baseline characteristics of included compared to excluded participants.**
(DOCX)

**S2 Table. Variation matrix of movement behaviors by sex.**
(DOCX)

**S3 Table. Variation matrix of movement behaviors by race/ethnicity.**
(DOCX)

**S4 Table. The association between activity composition (expressed as isometric log ratios) and percent weight change by sex and race/ethnicity.**
(DOCX)

**S5 Table. The association between activity composition (expressed as isometric log ratios) and weight change by sex and race/ethnicity among those with BMI $\geq$ 25 kg/m$^2$ at baseline.**
(DOCX)

## Acknowledgments

The authors express sincere appreciation to all Cancer Prevention Study-3 AVSS participants and ACS staff. The views expressed here are those of the authors and do not necessarily represent the American Cancer Society or the American Cancer Society–Cancer Action Network.

## Author Contributions

**Conceptualization:** Erika Rees-Punia, Ying Wang, Alpa V. Patel.

**Data curation:** Alpa V. Patel.

**Formal analysis:** Erika Rees-Punia.

**Investigation:** Erika Rees-Punia.

**Methodology:** Erika Rees-Punia, Mark A. Guinter, Ying Wang, Alpa V. Patel.

**Supervision:** Susan M. Gapstur, Alpa V. Patel.

**Writing – original draft:** Erika Rees-Punia.

**Writing – review & editing:** Erika Rees-Punia, Mark A. Guinter, Susan M. Gapstur, Ying Wang, Alpa V. Patel.

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
