## [Decision Letter · Decision Letter 0]

14 Sep 2020

PONE-D-20-22629

Composition of Time and Weight Change in Latinx, Black and White Participants

PLOS ONE

Dear Dr. Rees-Punia,

Thank you for submitting your manuscript to PLOS ONE. After careful consideration, we feel that it has merit but does not fully meet PLOS ONE’s publication criteria as it currently stands. Therefore, we invite you to submit a revised version of the manuscript that addresses the points raised during the review process.

We look forward to receiving your revised manuscript.

Kind regards,

Sze Yan Liu, PhD

Academic Editor

PLOS ONE

Journal Requirements:

2. Please refer to any sample size calculations performed prior to participant recruitment. If these were not performed please justify the reasons. Please refer to our statistical reporting guidelines for assistance (https://journals.plos.org/plosone/s/submission-guidelines.#loc-statistical-reporting).

3.Thank you for stating the following in the Funding Source Section of your manuscript:

[The American Cancer Society (ACS) funds the creation, maintenance, and updating of the Cancer Prevention Study-3.]

 [The author(s) received no specific funding for this work.]

4.We note that you have indicated that data from this study are available upon request. PLOS only allows data to be available upon request if there are legal or ethical restrictions on sharing data publicly. For information on unacceptable data access restrictions, please see http://journals.plos.org/plosone/s/data-availability#loc-unacceptable-data-access-restrictions.

Reviewers' comments:

Reviewer's Responses to Questions

**Comments to the Author**

1. Is the manuscript technically sound, and do the data support the conclusions?

Reviewer #1: Yes

Reviewer #2: Partly

Reviewer #3: Partly

2. Has the statistical analysis been performed appropriately and rigorously? 

Reviewer #1: No

Reviewer #2: I Don't Know

Reviewer #3: No

3. Have the authors made all data underlying the findings in their manuscript fully available?

Reviewer #1: No

Reviewer #2: No

Reviewer #3: Yes

4. Is the manuscript presented in an intelligible fashion and written in standard English?

Reviewer #1: Yes

Reviewer #2: Yes

Reviewer #3: Yes

5. Review Comments to the Author

Reviewer #1: Summary:

All in all this paper represents a novel application of compositional analyses. I am not aware of any other studies looking at the relationship between the composition of time spent in movement behaviours in a longitudinal fashion with repeated measures. Modelling weight gain is an interesting (albeit limited) application and the authors are correct in saying prospective associations are lacking given most applications to date are cross-sectional in nature. Also, presenting sex and ethnic stratification is another important aspect of this study.

That being said, the study has several major limitations and may have misapplied some of the statistical analyses. More data pertaining to the model itself and a larger part of the discussion section dedicated to discussing the threats to validity (mostly misclassification and selection bias) which limit the validity and generalizability of the findings.

Due to the novelty of the findings if these limitations are addressed more thoroughly and analyses corrected, I think this paper would make a good fit within the current literature.

Major Comments:

Methods:

Line 59-63 explain that the sample was stratified by sex and race/ethnicity. How were these sample sizes determined within each stratum? More detail is needed on how the sample size in each stratum was arrived at. This seems more like convenience sampling rather than stratified random sampling. I don’t see a sample size or power calculation reference anywhere? This is very important to delineate for a reader to judge the validity and generalizability of the sample.

Line67-68 Please show the data comparing included vs excluded participants (in a supplementary figure) so the reader can decide for themselves. By significantly do you mean not P-value <0.05? That is not a good criterion for deciding on potential missing data bias especially in subgroups like ethnicity/race

Line 73 Maybe I am confused but participants wore the device twice. Do you have two distinct period of wear time each of 7 days? If so, how did you choose which to include?

Results

I don’t see smoking in the tables. All variables stated in methods should be described in table 1 and 2

The F-Tests are not done correctly it appears. You present the F-test from the full model which is adjusted for covariates (as per the table legend). The F-Test refers to the SS of all covariates so the composition plus the covariates. These F-tests should either be for an unadjusted model (with only the composition) or a partial F-Test comparing the models with and without the compositional variables. You can only say here the full model is associated with weight change. It is entirely possible the composition on its own is not associated with the outcome and only appears to be when in the full model (of course sex and age among others will be related to weight change)

I don’t see the parameter estimates and hypothesis tests for the compositional variables (i.e. log ratio parameters). This is important to see which behaviours have significant contributions. Model parameter estimates should always be shown in the results section.

I am confused you used a 5-part composition (with VPA) but then all of your isotemporal substitutions were using MVPA? What happened to the focus and importance on VPA? Was the substitution stronger for VPA compared to MPA? Did you fit a separate 4-part composition model with MVPA? This seems inconsistent with modelling.

Discussion:

Biggest limitation is self-reported weight. This needs to be emphasized especially since this can lead to differential misclassification bias that may be directly related to the movement behaviors especially among the reported strata (sex or race/ethnicity). Please provide quantification of the correlation and whatever other metric of relevance demonstrate validity between self-reported and measured weight in your study. You need to sell the reader on this relationship and report the findings from unpublished work more thoroughly.

Weight change is only a small piece of the puzzle. The authors should elaborate on movement behaviours influencing health beyond just weight loss and this study only looks at this one aspect of health which limits its applicability and relevance.

Minor Comments:

-The title does not seem specific enough. Composition of Time can refer to almost anything. This paper is about Composition of Time Spent in Movement Behvaiours.

-line 49 subcohort may be more appropriate than nested cohort

-In the results section along with % day spent in each movement behaviour it would be good to show mean minutes as well or show the minutes in table 1 and 2 among the different strata. Obviously, leave SD blank for these.

-For table 1 don’t say average but put actual statistic which was the arithmetic mean it appears

-lines 225-226 seem out of place and not relevant quite yet

Reviewer #2: In the presented manuscript entitled Composition of Time and Weight Change in Latinx, Black and White Participants, the authors focus on elucidating the relationship between physical behavior and health risks, which are represented in the study by changes in body weight. More specifically, the authors assess the causal relationship between the way of spending time during the day in terms of physical behavior and the three-year prospective change in body weight in adults. For this purpose, they used objective (accelerometer-determined physical activity) and subjective data collection techniques (sleep duration, body weight). Data processing from accelerometers corresponds to the latest methodological trends, which significantly strengthens the quality of the study. In the study, the authors further evaluate the potential effect of reducing sedentary behavior in favor of either sleep, mild physical activity, moderate physical activity, or intense physical activity.

The study concludes with the thesis that the way of spending the time of daytime sitting, sleeping, or different intensities of physical activity is associated with a three-year change in body weight in women. They also emphasize the partial contribution of intense physical activity to the maintenance of body weight levels. He further states that the change of 30 minutes of sedentary sleep behavior is associated with a slight increase in weight in white men and women.

Although I find the study interesting, I have noticed several serious shortcomings/weaknesses in the manuscript.

1. The key issue of the study is the importance of long-term changes in body weight during life in adults. Without clarifying the significance of this somatic change in a broader context, the attractiveness of the presented study purpose is reduced. In the introduction section, the authors "only" point out to the fact that an inadequate approach has been used to assess the associations between movement-related behavior and changes in body weight. I perceive this clarification of the issue as incomplete.

2. The assessed association is based on a regression model, where it is assumed that the composition of time-use (movement-related) behavior is stable for three years. Given the plausibility of this assumption, this fact should be emphasized in the study and, above all, the results of the study must be discussed in this way. Although studies based on the same premise exist, I do not consider the model in the real environment to be adequate.

3. The study of the influence of movement-related behavior on health is complicated. The main reason is the breadth and complexity of both constructs. In a broader context, I consider the chosen regression model to be too simplified, as it does not take into account other important variables. In this sense, I consider the overall concept of the study to be inadequate.

4. Considering that weight is one of the two most important variables of the study, I perceive the chosen method of data collection (self-report) on body weight as a fundamental weakness of the study. Bias related to this method is well described in the literature. This fact is not taken into account at all in the study. Moreover, the chosen categorization of body weight changes (i.e. weight loss, gain, and maintenance) only further reduces the overall validity of the study.

5. The methods are not described enough to enable the study replicability. The authors inform that the participants coded their daily time-use behaviors in 15-minute epochs concurrently wearing accelerometers. It is unclear how the authors processed this data with the accelerometer data. Furthermore, I miss more detailed information about the estimation of the time spent in the mentioned behaviors such as sedentary, light physical activity, moderate physical activity, and vigorous physical activity. It is only stated what techniques were used (i.e. sojourn-3 axis algorithm, hybrid machine-learning, neural network, and decision tree analysis algorithm) which is insufficient. Similarly, more information regarding the data processing from accelerometers, isometric log-ratio transformation model, and regression model settings are needed.

6. The study lacks an explanation of why the authors evaluate a specific substitution of sedentary behavior and consequently rationale for the choice of 30 minutes. For example, why are not also sleep substitution or light physical activity substitution tested? Why were not substitutions lasting 5, 10, or 15 minutes analyzed?

7. I do not consider the results presented sufficiently (see Submission Guidelines; Lang T, Altman D. Statistical Analyses and Methods in the Published Literature: the SAMPL Guidelines).

Reviewer #3: Overall points

1. There is a mismatch between how the rationale was described and how the analysis has been performed. The rationale does not cover any points about why it is important to look into gender and ethnicity differences while later in the analysis, these differences were explored. Seems like researchers did not have research questions on these differences before but later found these differences important to understand. Suggest the rationale should cover these differences. So suggest to add rationale for why is it important to stratify results on sex and ethnicity?

2. Keeping in mind this, it is also important to explore possible interaction with ethnicity and gender in the study (this is not done yet in the intro)

3. The study is about understanding a whole composition of time use and not just about understanding the association of sedentary time relative to other behaviors with weight change, as shown in the iso sub models in the study. Suggest the results should be presented for other reallocations as well. What is the reason that authors explored iso sub models for replacements between sedentary time and other behaviors and not between other behaviors? I think keeping in mind coda thinking, it is important to explore all possible combinations. It may be so that it is important to explore what if we replace LPA with MVPA or LPA with sleep.

4. The formation of the composition has some problems. First the composition contains sedentary, LPA, MPA and VPA. Authors try to incorporate this composition (in log form) in the model. But later in the iso sub model, authors combine MPA and VPA. Why is that? If authors wanted to combine MPA and VPA later, then authors should do that from the very top. As far as I understand this will violate some statistical rules. For example the ratios between parts would be different in both cases. Thus suggest to make your composition where authors have MVPA instead of MPA and VPA together with other parts of the composition.

5. In the study, there are lot of sub groups analysis, gender stratified and ethnicity stratified and a combination of the two. However, due to sample size limitation, authors could not run all these analysis. I suggest to keep the sub group analyses only to ethnicity and gender and not the combination of the two as authors do not have power for it and thus authors would not be able to properly explore this in the study.

6. Overall the manuscript could have been simple actually: understanding the prospective association between the whole composition and weight change, stratified on gender and ethnicity.

7. Also a major part of the results was dedicated on stratifying the weight change in three categories. So why authors used linear regression and not logit models where your outcome is in three categories. Authors might have an intention of using categories to further understand the results but this intention is not clear. Suggest to align analyses and results.

8. Results on the whole population should anyway be presented (at least in supplementary files), as a favor to meta analyses studies in the future.

Some other specific comments are here:

In introduction:

“Research supports that physical activity,[1] sedentary behavior,[2, 3] and sleep [4] may be

31 independently associated with the development of overweight or obesity”

Suggest to revise something in line with the fact the previous research has ‘tried’ investing the ‘independent’ effect of each single physical behaviors such as (…) with weight.

Statistical analysis

I am quite surprised that authors did not have any zeros in VPA when your average was 8 (and only 6 minutes in black ethnicity) minutes only. What is the range of this variable for different groups?

Page 6 line 115. I don’t understand what authors mean by ‘when feasible’. In any case authors would have very small groups to make stratification both on sex and ethnicity. Important to define right here what authors mean by ‘feasible’

Page 6 line 119. Suggest to write in line with that the iso sub models were used to interpret the estimates (that were in log form) obtained from multiple linear regressions instead of ‘To better understand which parts of the time-use composition were important for weight change’

What is the reason behind using only one to one reallocation? Not one to many?

Why would authors do a sensitivity analysis where authors restrict the analysis to overweight or obese participants at baseline? Usually it should be other way round where authors restrict the analysis on those without obesity and then follow them up to see if they develop obesity.

Percentage change in weight is a sensitivity analysis actually.

Adjusting for total energy expenditure (EE) is good but was this variable highly correlated with any other variables in the model? Leading to multicollinearity issue? Good to report the highest correlation of EE with any variable in the model.

Were models checked for statistical assumptions violations?

Are authors not using other r packages as well? Report other packages as well. Please add citations for those packages.

Suggest to change the unit of weight from lbs to kilos, unless stated by plos one.

There are no details on how authors measured confounders? Also need references for when authors say that authors selected confounders a priori.

Important that authors bring supplementary table 1 and 2 as main tables. Those are important numbers to understand the testing sample.

Did authors test for interaction for sex and ethnicity?

N for each sub group needs to be defined somewhere clearly, say in your descriptive tables.

Suggest to remove results of joint association. Authors do not have enough sample size to really explore this (for example authors could only do it for whites and not other ethnicities)

What is rationale for separating vpa keeping in mind it is such a short duration.

Discussion

Line 216: “Analyses further suggest that time in VPA may be important, relative to other

behaviors, for weight maintenance.” i am not sure which analysis support this result? Your linear reg is your main analysis. That does not explore this association .

also authors are now using MVPA not MPA and VPA?

I did not go further in the discussion because it will change if authors decide to bring other estimates from other reallocations in the manuscript.

6. PLOS authors have the option to publish the peer review history of their article (what does this mean?). If published, this will include your full peer review and any attached files.

Reviewer #1: **Yes: **Robert Talarico

Reviewer #2: No

Reviewer #3: No

---

## [Author Response · Author response to Decision Letter 0]

6 Nov 2020

Comments to the Author

Reviewer #1

All in all this paper represents a novel application of compositional analyses. I am not aware of any other studies looking at the relationship between the composition of time spent in movement behaviours in a longitudinal fashion with repeated measures. Modelling weight gain is an interesting (albeit limited) application and the authors are correct in saying prospective associations are lacking given most applications to date are cross-sectional in nature. Also, presenting sex and ethnic stratification is another important aspect of this study.

That being said, the study has several major limitations and may have misapplied some of the statistical analyses. More data pertaining to the model itself and a larger part of the discussion section dedicated to discussing the threats to validity (mostly misclassification and selection bias) which limit the validity and generalizability of the findings.

Due to the novelty of the findings if these limitations are addressed more thoroughly and analyses corrected, I think this paper would make a good fit within the current literature.

• Response: Thank you for your interest and for taking the time to provide a thoughtful review. 

Methods:

Line 59-63 explain that the sample was stratified by sex and race/ethnicity. How were these sample sizes determined within each stratum? More detail is needed on how the sample size in each stratum was arrived at. This seems more like convenience sampling rather than stratified random sampling. I don’t see a sample size or power calculation reference anywhere? This is very important to delineate for a reader to judge the validity and generalizability of the sample.

• Response: We have added a bit more detail around the AVSS sub-sample, including two references that discuss the study design in more detail: “The original intent of the AVSS was to validate the physical activity and sedentary time questionnaire in multiple racial/ethnic groups, therefore the first 300 non-Latina white female, 150 non-Latino white male, 150 Latinx, and 150 non-Latinx African-American/Black participants to complete the routine 2015 CPS-3 follow-up survey were enrolled into the AVSS (AVSS discussed in more detail elsewhere).”

Line67-68 Please show the data comparing included vs excluded participants (in a supplementary figure) so the reader can decide for themselves. By significantly do you mean not P-value <0.05? That is not a good criterion for deciding on potential missing data bias especially in subgroups like ethnicity/race

• Response: We have added an additional supplementary table (Supp Table 1) comparing included vs. excluded participants. 

Line 73 Maybe I am confused but participants wore the device twice. Do you have two distinct period of wear time each of 7 days? If so, how did you choose which to include?

• Response: Participants indeed wore the device twice- during two non-consecutive quarters- for seven days each quarter. We used all valid days from both quarters. We have clarified this point by adding the following, “During two non-consecutive quarters (Q1/Q3 or Q2/Q4), participants wore an Actigraph GT3x+ accelerometer on the hip aligning with the midline of the non-dominant thigh. Participants were instructed to wear the device for seven consecutive days during all waking hours, except when bathing or swimming during both of their assigned quarters (i.e., seven days in Q1/seven days in Q3 OR seven days in Q2/seven days in Q4).”

Results

I don’t see smoking in the tables. All variables stated in methods should be described in table 1 and 2

• Response: We have added smoking to Table 1, though we still are not including it in our models given low prevalence (previously on Lines 129-130: “Smoking status information was available, however very few participants smoked (2% current smokers), so smoking status was not included in the models”). 

The F-Tests are not done correctly it appears. You present the F-test from the full model which is adjusted for covariates (as per the table legend). The F-Test refers to the SS of all covariates so the composition plus the covariates. These F-tests should either be for an unadjusted model (with only the composition) or a partial F-Test comparing the models with and without the compositional variables. You can only say here the full model is associated with weight change. It is entirely possible the composition on its own is not associated with the outcome and only appears to be when in the full model (of course sex and age among others will be related to weight change)

• Response: The results presented in Table 3 are actually from the Wald chi square ANOVA type II test of the linear regression model. We have clarified this detail in-text and added it as a footnote to Table 3, “The geometric mean of time spent in each time-use behavior and the log-ratio variances between each pair of behaviors was calculated. For the main analyses, separate multiple linear regression models were first used to examine associations between the 5-part time-use composition (expressed as isometric log-ratios) and three-year weight change (absolute change and percent change modeled continuously) stratified by sex or race/ethnicity and, where feasible, by sex and race/ethnicity. Models were checked to ensure the assumptions of linearity, normality, homoskedasticity and leverage were not violated. A Wald chi square ANOVA type II test of the multiple linear regression models was used to assess the significance of the 5-part time use composition.”

In other words, these F, SS, and p values are specifically for the activity composition (expressed as ilr) in a Wald Chi square ANOVA model that is adjusted for covariates. 

I don’t see the parameter estimates and hypothesis tests for the compositional variables (i.e. log ratio parameters). This is important to see which behaviours have significant contributions. Model parameter estimates should always be shown in the results section.

• Response: We used multiple linear regression models to first explore the association between three-year weight change (continuously) and the activity composition (expressed as isometric log ratios) as the explanatory variable. This regression model will provide one beta estimate and p-value for each ilr (here, there are 4 ilr, as we have a 5-part composition). However, these values are not necessarily interpretable in an intuitive way. According to Chastin, 2015, “… the model p-value is an indicator of whether or not the composition has a significant association with the outcome Y. In standard linear regression we normally interpret each β as the strength of the association between the behavior and Y. This is often used to understand if time spent in a specific behavior has an independent effect on Y and to quantify this potential effect. As discussed above, because of the compositional nature of time spent in physical activity behavior, this way of thinking in largely nonsensical and should be abandoned. Any conclusion drawn from this approach is likely to have limited trustworthiness. Instead we should reason in terms of relative amount spent in one behavior with respect to the others.” Similarly, Dumid (2017, 2018) states, “components of time-use data should not be interpreted in isolation from the remaining parts, as all parts are necessarily related to each other”. Thus, as a next step, we used a Wald chi square ANOVA type II test of the multiple linear regression models to assess the significance of the 5-part time use composition.

While it is certainly of interest to determine whether the time-use composition is a significant predictor of weight change, we indeed agree that it is equally important to discover which time-use components are driving the relationship. According to Dumid, 2018 (based on Aitchinson geometry), this can be tackled in two main ways: “First, an interpretable form of ilr can be used, and the interpretable coordinate’s regression estimate and p-value be considered. Second, the linear model can be used to estimate how a health outcome changes when absolute durations of time are reallocated between time-use components.” Although both of these ways are correct, we feel that the first option can be a bit more difficult to interpret as, “the values reported must be interpreted with direct reference to the corresponding ilr coordinate. However, the numerical value associated with the beta coefficients cannot be interpreted in absolute terms, as it corresponds to a relative construct. Consequently, the beta coefficients must be interpreted from a reference composition.” Therefore, we followed the second method by presenting time substitution analyses, as we feel this is much more translatable to a public health audience.

Note that this method is followed in many published papers of compositional analyses of movement time use. Each of the following papers followed this method, and none of them show parameter estimates for individual behaviors:

o Dumuid D, Lewis LK, Olds TS, Maher C, Bondarenko C, Norton L. Relationships between older adults' use of time and cardio-respiratory fitness, obesity and cardio-metabolic risk: A compositional isotemporal substitution analysis. Maturitas. 2018;110:104-110.

o Fairclough SJ, Dumuid D, Mackintosh KA, et al. Adiposity, fitness, health-related quality of life and the reallocation of time between children's school day activity behaviours: A compositional data analysis. Preventive medicine reports. 2018;11:254-261.

o Fairclough SJ, Dumuid D, Taylor S, et al. Fitness, fatness and the reallocation of time between children's daily movement behaviours: an analysis of compositional data. The international journal of behavioral nutrition and physical activity. 2017;14(1):64.

o Gupta N, Dumuid D, Korshoj M, Jorgensen MB, Sogaard K, Holtermann A. Is Daily Composition of Movement Behaviors Related to Blood Pressure in Working Adults? Medicine and science in sports and exercise. 2018;50(10):2150-2155.

o Dorothea Dumuid, Melissa Wake, Susan Clifford, David Burgner, John B. Carlin, Fiona K. Mensah, François Fraysse, Kate Lycett, Louise Baur, Timothy Olds. The Association of the Body Composition of Children with 24-Hour Activity Composition, The Journal of Pediatrics, Volume 208, 2019, Pages 43-49.e9, ISSN 0022-3476

I am confused you used a 5-part composition (with VPA) but then all of your isotemporal substitutions were using MVPA? What happened to the focus and importance on VPA? Was the substitution stronger for VPA compared to MPA? Did you fit a separate 4-part composition model with MVPA? This seems inconsistent with modelling.

• Response: We were able to use a 5-part composition for the main analyses, but unfortunately needed to use a 4-part composition for the isotemporal substitution analyses. We added detail to explain why in the methods: “Compositional isotemporal substitution analyses explored the replacement of sedentary time with sleep, LPA, or moderate-vigorous physical activity combined (as average time in VPA alone was well under 30 min/day (mean: 13 min/day) and would therefore result in negative time when modeling 30 minutes of replacement). 

Based on the three main principles of compositional analyses, this should not violate any assumptions; according to Dumid, 2017, “The third principle is sub-compositional coherence. The results from a subset of compositional parts must not be reliant on any other parts that are excluded from the subset. In other words, the relative difference between two parts must be the same, regardless of how many parts are included in the analysis.”

Discussion:

Biggest limitation is self-reported weight. This needs to be emphasized especially since this can lead to differential misclassification bias that may be directly related to the movement behaviors especially among the reported strata (sex or race/ethnicity). Please provide quantification of the correlation and whatever other metric of relevance demonstrate validity between self-reported and measured weight in your study. You need to sell the reader on this relationship and report the findings from unpublished work more thoroughly.

• Response: The validation study has been published since submitting the current paper, so we are able to add a bit more detail on this: “Finally, measurement error in self-reporting weight may also explain these findings, though a recent validation study comparing self-reported weight to measured weight among 2,643 CPS-3 participants found that reported weight was highly correlated with measured weight among men (Pearson r = 0.98) and women (Pearson r = 0.99). Pertinent to the current study, the validation study also found that mean differences in self-reported compared to measured weight did not differ by race/ethnicity among men (p for difference = 0.528) or women (p = 0.122).”

Weight change is only a small piece of the puzzle. The authors should elaborate on movement behaviours influencing health beyond just weight loss and this study only looks at this one aspect of health which limits its applicability and relevance.

• Response: Thank you for bringing up this excellent point- weight maintenance/weight change is only one potential benefit of a healthful balance of movement behaviors. We have added detail on this in the limitations section: “Additionally, this study focused on only one potential benefit of a healthful balance of time in movement behaviors; it is important to note that movement behaviors likely influence health in many ways beyond weight maintenance or change.”

Minor Comments:

-The title does not seem specific enough. Composition of Time can refer to almost anything. This paper is about Composition of Time Spent in Movement Behvaiours.

• Response: We have updated the title to: “Composition of Time in Movement Behaviors and Weight Change in Latinx, Black and White Participants”.

-line 49 subcohort may be more appropriate than nested cohort

• Response: We have updated this to read “…in a sub-study of the Cancer Prevention Study-3…”.

-In the results section along with % day spent in each movement behaviour it would be good to show mean minutes as well or show the minutes in table 1 and 2 among the different strata. Obviously, leave SD blank for these.

• Response: We have added mean daily minutes spent in each behavior: “Overall, participants spent 41.6% of the day sedentary (599 min/day), 18.4% in LPA (264 min/day), 4.0% in MPA (57 min/day), 0.6% in VPA (9 min/day), and 35.5% sleeping (511 min/day) on average.”

-For table 1 don’t say average but put actual statistic which was the arithmetic mean it appears

• Response: We have updated this to, “Arithmetic Mean”. 

-lines 225-226 seem out of place and not relevant quite yet

• Response: We have edited this section, including this statement: “Other compositional analyses, although cross-sectional in nature, have also reported significant associations between the activity composition and weight in adults. There are also several cross-sectional compositional studies of the relationship between movement behaviors and weight or adiposity in children and adolescents. One important difference between the current study and the existing literature, aside from the cross-sectional versus longitudinal designs, is that prior studies to date have not been adequately powered to assess the associations by race/ethnicity. Although studies of the composition of time in movement behaviors are relatively sparse, the 24-hour movement framework is more widely recognized, as is evident in the national physical activity guidelines for Canada and Australia.”

Reviewer #2: 

In the presented manuscript entitled Composition of Time and Weight Change in Latinx, Black and White Participants, the authors focus on elucidating the relationship between physical behavior and health risks, which are represented in the study by changes in body weight. More specifically, the authors assess the causal relationship between the way of spending time during the day in terms of physical behavior and the three-year prospective change in body weight in adults. For this purpose, they used objective (accelerometer-determined physical activity) and subjective data collection techniques (sleep duration, body weight). Data processing from accelerometers corresponds to the latest methodological trends, which significantly strengthens the quality of the study. In the study, the authors further evaluate the potential effect of reducing sedentary behavior in favor of either sleep, mild physical activity, moderate physical activity, or intense physical activity.

The study concludes with the thesis that the way of spending the time of daytime sitting, sleeping, or different intensities of physical activity is associated with a three-year change in body weight in women. They also emphasize the partial contribution of intense physical activity to the maintenance of body weight levels. He further states that the change of 30 minutes of sedentary sleep behavior is associated with a slight increase in weight in white men and women. Although I find the study interesting, I have noticed several serious shortcomings/weaknesses in the manuscript.

• Response: Thank you for taking the time to review our work. 

1. The key issue of the study is the importance of long-term changes in body weight during life in adults. Without clarifying the significance of this somatic change in a broader context, the attractiveness of the presented study purpose is reduced. In the introduction section, the authors "only" point out to the fact that an inadequate approach has been used to assess the associations between movement-related behavior and changes in body weight. I perceive this clarification of the issue as incomplete.

• Response: We believe the reviewer is expressing concern about the importance of longer-term weight change (i.e., longer than 3 years). We agree that more longitudinal studies (including studies of weight change over a longer period of time) would be very beneficial and have included this statement in our conclusions: “However, more longitudinal studies using compositional analyses are needed to understand the complex relationships between time-use behaviors and weight change.” 

Importantly, all other existing studies of the composition of time and weight are cross-sectional, thus we feel these analyses make an important contribution as the first compositional study of time-use behaviors with prospectively collected, longitudinal weight change.

2. The assessed association is based on a regression model, where it is assumed that the composition of time-use (movement-related) behavior is stable for three years. Given the plausibility of this assumption, this fact should be emphasized in the study and, above all, the results of the study must be discussed in this way. Although studies based on the same premise exist, I do not consider the model in the real environment to be adequate.

• Response: We had two time points of accelerometer data over a one-year period, a point we have clarified here: “Data collection for the CPS-3 AVSS occurred over a one-year period (2015), which was split into four quarters. During two non-consecutive quarters (Q1/Q3 or Q2/Q4), participants wore an Actigraph GT3x+ accelerometer on the hip aligning with the midline of the non-dominant thigh. Participants were instructed to wear the device for seven consecutive days during all waking hours, except when bathing or swimming, during both assigned quarters (i.e., seven days in Q1/seven days in Q3 OR seven days in Q2/seven days in Q4).” 

We found that Q1/2 vs. Q3/4 data were very stable for most participants (see table below). Therefore, the assumption that movement behaviors were likely relatively stable for years 2 and 3 does not appear to be a limitation.

 Q1/Q2 Q3/Q4

Mean sedentary time (min.) 591 584

Mean LPA (min.) 272 260

Mean MVPA (min.) 68 74

3. The study of the influence of movement-related behavior on health is complicated. The main reason is the breadth and complexity of both constructs. In a broader context, I consider the chosen regression model to be too simplified, as it does not take into account other important variables. In this sense, I consider the overall concept of the study to be inadequate.

• Response: We agree that the study of movement behaviors and weight is indeed complex. One strength of this study is the accelerometer-assessed sedentary time, LPA, and MPVA, especially as our participants had very high compliance with our strict wear protocol of 14 hours/day. We were also able to include covariates likely associated with weight in our models, including sex, race/ethnicity, age, average daily caloric intake, comorbidity score, height, and smoking status. Another covariate potentially important for weight change may include diet quality, which was unfortunately unavailable- a point that we now discuss in detail: “Alternatively, diet may have played a large role in participant weight loss. All models were adjusted for energy intake estimated from a food frequency questionnaire (FFQ), but some studies suggest that FFQs may not accurately estimate total energy intake. were also unable to account for diet change during this period. Some studies suggest that while physical activity may not independently contribute to significant weight loss without accompanying dietary changes, it may play a more substantive role in weight maintenance. Thus, it is possible that there may be a role of time-use behaviors in weight maintenance, however dietary changes are needed to elicit significant weight loss.” …”As previously noted, another limitation was the limited dietary data.”

4. Considering that weight is one of the two most important variables of the study, I perceive the chosen method of data collection (self-report) on body weight as a fundamental weakness of the study. Bias related to this method is well described in the literature. This fact is not taken into account at all in the study. Moreover, the chosen categorization of body weight changes (i.e. weight loss, gain, and maintenance) only further reduces the overall validity of the study.

• Response: Our group recently published a validation study of self-reported weight in CPS-3. We have expanded on the potential biases associated with self-reported weight, and added some detail from the recent validation study to justify the use of this method in CPS-3: “Finally, error in self-reporting weight may also explain these findings, though a recent validation study comparing self-reported weight to measured weight among 2,643 CPS-3 participants found that reported weight was highly correlated with measured weight among men (Pearson r = 0.98) and women (Pearson r = 0.99). Pertinent to the current study, the validation study also found that mean differences in self-reported compared to measured weight did not differ by race/ethnicity among men (p for difference = 0.528) or women (p = 0.122).” Nevertheless, we understand this concern and have therefore still included the use of self-reported weight as a primarily limitation of this study in the discussion section. 

5. The methods are not described enough to enable the study replicability. The authors inform that the participants coded their daily time-use behaviors in 15-minute epochs concurrently wearing accelerometers. It is unclear how the authors processed this data with the accelerometer data. Furthermore, I miss more detailed information about the estimation of the time spent in the mentioned behaviors such as sedentary, light physical activity, moderate physical activity, and vigorous physical activity. It is only stated what techniques were used (i.e. sojourn-3 axis algorithm, hybrid machine-learning, neural network, and decision tree analysis algorithm) which is insufficient. Similarly, more information regarding the data processing from accelerometers, isometric log-ratio transformation model, and regression model settings are needed.

• Response: We have clarified our use of diary data and the Choi/Sojourn algorithms, including references: “Actigraph data were processed using the Choi algorithm to calculate accelerometer wear time. Days with less than 14 hours of wear time per day were not considered valid and excluded from analyses, as they were unlikely to represent a full waking period. The sojourn-3 axis algorithm, which is a hybrid machine-learning, neural network, and decision tree analysis algorithm, was used to estimate daily sedentary, light intensity physical activity (LPA), moderate intensity physical activity (MPA), and vigorous intensity physical activity (VPA) time. Participants were instructed to remove the accelerometers before bed, thus, time spent sleeping was estimated from 24-hour diaries. Participants were asked to code their daily time-use behaviors, including sleep, in 15-minute epochs for the seven days concurrent with accelerometer wear. Average time spent sleeping was calculated from the diaries for participants with at least three valid days of diary data concurrent with accelerometer wear. Diary data was not otherwise used for sedentary time, LPA, MPA, or VPA, as accelerometer-assessed data were available.”

We have also added detail around covariates included in the model: “All models were sex and/or race/ethnicity stratified, and the following covariates from 2015 CPS-3 surveys were identified a priori: age, race/ethnicity (White, Black, Latinx; for sex stratified models), sex (for race/ethnicity stratified models), average daily caloric intake (continuous; estimated from the food frequency questionnaire), comorbidity score (sum of comorbidities including hypertension, hypercholesterolemia, and type 2 diabetes), and height (continuous).”

6. The study lacks an explanation of why the authors evaluate a specific substitution of sedentary behavior and consequently rationale for the choice of 30 minutes. For example, why are not also sleep substitution or light physical activity substitution tested? Why were not substitutions lasting 5, 10, or 15 minutes analyzed?

• Response: We now present some additional substitutions in Table 4. 

Table 4. Estimated difference in 3-year weight change associated with reallocation of 30 minutes 

30 min sedentary time per day reallocated to: Estimated difference (95% confidence interval), lbs.

All women 

 30 min LPA -0.20 lbs. (-0.78, 0.38)

 30 min MVPA -0.71 lbs. (-2.74, 1.31)

 30 min sleep 0.82 lbs. (-0.06, 1.69)

White women 

 30 min LPA -0.80 lbs. (-1.61, 0.02)

 30 min MVPA -0.39 lbs. (-2.90, 2.11)

 30 min sleep 1.52 lbs. (0.25, 2.79)

White men 

 30 min LPA -0.20 lbs. (-0.93, 0.53)

 30 min MVPA 0.62 lbs. (-1.64, 2.89)

 30 min sleep 1.31 lbs. (0.40, 2.21)

Latinx 

 30 min LPA 0.28 lbs. (-0.42, 0.99)

 30 min MVPA -3.49 lbs. (-6.76, -0.22)

 30 min sleep -1.14 lbs. (-2.29, 0.02)

30 min sleep time per day 

reallocated to: Estimated difference (95% confidence interval), lbs.

All women 

 30 min LPA -0.98 lbs. (-1.99, 0.03)

 30 min MVPA -1.50 lbs. (-3.54, 0.04)

 30 min sedentary -0.79 lbs. (-1.66, 0.08)

White women 

 30 min LPA -2.26 lbs. (-3.73, -0.78)

 30 min MVPA -1.85 lbs. (-4.42, -0.72)

 30 min sedentary -1.48 lbs. (-2.73, 0.23)

White men 

 30 min LPA -1.46 lbs. (-2.64, -0.27)

 30 min MVPA -0.63 lbs. (-2.94, 0.67)

 30 min sedentary -1.28 lbs. (-2.16, 0.39)

Latinx 

 30 min LPA 1.38 lbs. (-0.09, 2.67)

 30 min MVPA -2.40 lbs. (-5.60, -0.08)

 30 min sedentary 1.12 lbs. (-0.02, 2.26)

30 min LPA per day 

reallocated to: Estimated difference (95% confidence interval), lbs.

All women 

 30 min sleep 0.97 lbs. (-0.01, 1.96)

 30 min MVPA -0.56 lbs. (-2.68, 1.56)

 30 min sedentary 0.14 lbs. (-0.39, 0.69)

White women 

 30 min sleep 2.20 lbs. (0.76, 3.64)

 30 min MVPA -0.28 lbs. (-2.38, 2.94)

 30 min sedentary 0.66 lbs. (-0.10, 1.41)

White men 

 30 min sleep 1.43 lbs. (0.29, 2.59)

 30 min MVPA 0.75 lbs. (-1.80, 3.30)

 30 min sedentary 0.10 lbs. (-0.55, 0.76)

Latinx 

 30 min sleep -1.34 lbs. (-2.61, -0.08)

 30 min MVPA -3.70 lbs. (-6.76, -0.22)

 30 min sedentary 0.18 lbs. (-0.84, 0.48)

Composition of time not associated with men or Black participants, therefore all women, white women, white men, and Latinx participants are included in compositional isotemporal substitution models. All models are adjusted for age, race/ethnicity or sex, average kcal/day (FFQ estimate), comorbidity score, and height.

We chose 30 minute substitutions as 1) 30 minutes/day of PA is roughly consistent with the US physical activity guidelines and 2) 30 minutes/day is a large enough change in activity to potentially elicit a change in weight, yet it still a manageable change from a behavioral perspective (compared to 60 minutes, 90 minutes, etc.).

7. I do not consider the results presented sufficiently (see Submission Guidelines; Lang T, Altman D. Statistical Analyses and Methods in the Published Literature: the SAMPL Guidelines).

• Response: We have added considerable detail throughout the results, including updated in-text results and tables (highlighted throughout). 

Reviewer #3: Overall points

1. There is a mismatch between how the rationale was described and how the analysis has been performed. The rationale does not cover any points about why it is important to look into gender and ethnicity differences while later in the analysis, these differences were explored. Seems like researchers did not have research questions on these differences before but later found these differences important to understand. Suggest the rationale should cover these differences. So suggest to add rationale for why is it important to stratify results on sex and ethnicity?

• Response: We have added rationale for the importance of stratifying on sex and race/ethnicity in the introduction.

2. Keeping in mind this, it is also important to explore possible interaction with ethnicity and gender in the study (this is not done yet in the intro)

• Response: We agree that it is important to explore race/ethnicity, sex, and the interaction between the two. We did indeed explore interaction of race/ethnicity and sex, though we had too few Latinos (n=36) and Black males (n=41) to show fully stratified analyses. 

3. The study is about understanding a whole composition of time use and not just about understanding the association of sedentary time relative to other behaviors with weight change, as shown in the iso sub models in the study. Suggest the results should be presented for other reallocations as well. What is the reason that authors explored iso sub models for replacements between sedentary time and other behaviors and not between other behaviors? I think keeping in mind coda thinking, it is important to explore all possible combinations. It may be so that it is important to explore what if we replace LPA with MVPA or LPA with sleep.

• Response: We appreciate this suggestion and have added additional substitutions in Table 4.

4. The formation of the composition has some problems. First the composition contains sedentary, LPA, MPA and VPA. Authors try to incorporate this composition (in log form) in the model. But later in the iso sub model, authors combine MPA and VPA. Why is that? If authors wanted to combine MPA and VPA later, then authors should do that from the very top. As far as I understand this will violate some statistical rules. For example the ratios between parts would be different in both cases. Thus suggest to make your composition where authors have MVPA instead of MPA and VPA together with other parts of the composition.

• Response: We were able to use a 5-part composition for the main analyses, but unfortunately needed to use a 4-part composition for the isotemporal substitution analyses. We added detail to explain why in the methods: “Compositional isotemporal substitution analyses explored the replacement of sedentary time with sleep, LPA, or moderate-vigorous physical activity combined (as average time in VPA alone was well under 30 min/day (mean: 13 min/day) and would therefore result in negative time when modeling 30 minutes of replacement). 

Based on the three main principles of compositional analyses, this should not violate any assumptions; according to Dumid, 2017, “The third principle is sub-compositional coherence. The results from a subset of compositional parts must not be reliant on any other parts that are excluded from the subset. In other words, the relative difference between two parts must be the same, regardless of how many parts are included in the analysis.”

5. In the study, there are lot of sub groups analysis, gender stratified and ethnicity stratified and a combination of the two. However, due to sample size limitation, authors could not run all these analysis. I suggest to keep the sub group analyses only to ethnicity and gender and not the combination of the two as authors do not have power for it and thus authors would not be able to properly explore this in the study.

• Response: We agree it is unfortunate that our sample size is insufficient for joint analyses among Latinx and Black men and women, we do feel it is important to present the joint analyses we are able to present, as we did indeed see a significant interaction by race/ethnicity and sex (p = 0.03).

6. Overall the manuscript could have been simple actually: understanding the prospective association between the whole composition and weight change, stratified on gender and ethnicity.

• Response: That was statement conveys our overall goal. We hope this is conveyed more clearly in the manuscript with all the changes incorporated. 

7. Also a major part of the results was dedicated on stratifying the weight change in three categories. So why authors used linear regression and not logit models where your outcome is in three categories. Authors might have an intention of using categories to further understand the results but this intention is not clear. Suggest to align analyses and results.

• Response: We have clarified that our main analyses explored change in weight continuously, “For the main analyses, separate multiple linear regression models were first used to examine associations between the 5-part time-use composition (expressed as isometric log-ratios) and three-year weight change (absolute change and percent change modeled continuously) stratified by sex or race/ethnicity and, where feasible, by sex and race/ethnicity. Models were checked to ensure the assumptions of linearity, normality, homoskedasticity and leverage were not violated. A Wald chi square ANOVA type II test of the multiple linear regression models was used to assess the significance of the 5-part time use composition.” These results are shown in Table 3. 

The three categories of weight change were only used to compare the relative importance of time spent in each behavior between the three weight change groups- as shown in the Figures. 

Introduction:

“Research supports that physical activity,[1] sedentary behavior,[2, 3] and sleep [4] may be

independently associated with the development of overweight or obesity”

Suggest to revise something in line with the fact the previous research has ‘tried’ investing the ‘independent’ effect of each single physical behaviors such as (…) with weight.

• Response: We put the word “independently” in quotes to better set-up the following sentence discussing the co-dependent manner of these activities. 

Statistical analysis

I am quite surprised that authors did not have any zeros in VPA when your average was 8 (and only 6 minutes in black ethnicity) minutes only. What is the range of this variable for different groups?

• Response: We have added the range of values for this fairly young, active group: “To address the potential role of all movement behaviors (especially VPA) in weight loss a 5-part composition was considered for the main analysis; this was possible as the average daily VPA was at least 0.5 min/day for all included participants (range: 0.51-91.7 min/day).”

Page 6 line 115. I don’t understand what authors mean by ‘when feasible’. In any case authors would have very small groups to make stratification both on sex and ethnicity. Important to define right here what authors mean by ‘feasible’

• Response: By ‘when feasible’, we mean where the sample size is large enough to allow a sub-group analysis. We have updated this to read “stratified by sex or race/ethnicity and, where sample size allowed, by both sex and race/ethnicity.”

Page 6 line 119. Suggest to write in line with that the iso sub models were used to interpret the estimates (that were in log form) obtained from multiple linear regressions instead of ‘To better understand which parts of the time-use composition were important for weight change’

• Response: This has been updated to read, “Compositional isotemporal substitution analysis was then applied to quantify the change in weight associated with the reallocation of 30 minutes of sedentary time to an equivalent amount of time in another behavior, while keeping time spent in the remaining behaviors constant.[24]”

What is the reason behind using only one to one reallocation? Not one to many?

• Response: We presented one to one reallocations as we feel this is a bit more interpretable and translatable for a public health audience than a one to many reallocation. 

Why would authors do a sensitivity analysis where authors restrict the analysis to overweight or obese participants at baseline? Usually it should be other way round where authors restrict the analysis on those without obesity and then follow them up to see if they develop obesity.

• Response: We have added a statement on our rationale behind this sensitivity analysis: “…sensitivity analysis restricting to participants who were overweight or obese at baseline was also conducted, as normal weight participants may be less likely to experience a change in weight over a three-year period…”

Percentage change in weight is a sensitivity analysis actually.

• Response: That is correct. The percent change in weight results are shown in Supplemental Table 3. 

Adjusting for total energy expenditure (EE) is good but was this variable highly correlated with any other variables in the model? Leading to multicollinearity issue? Good to report the highest correlation of EE with any variable in the model.

• Response: We adjusted for energy intake (EI) in our models. EI was not highly correlated with any other covariates:

Pearson correlation coefficients 

 EI Height Age Comorbidity score

EI 1.0 -0.005 -0.031 0.041

Height -0.005 1.0 0.064 0.065

Age -0.031 0.064 1.0 0.311

Comorbidity score 0.041 0.065 0.311 1.0

 We have added a statement that models did not violate statistical assumptions. 

Were models checked for statistical assumptions violations?

• Response: Yes, we have added a statement clarifying that models did not violate statistical assumptions. 

Are authors not using other r packages as well? Report other packages as well. Please add citations for those packages.

• Response: “Compositions” is the only R package required for this analysis. 

There are no details on how authors measured confounders? 

• Response: We have added detail on where the confounders were measured, and the scale on which they were measured, “All models were sex and/or race/ethnicity stratified, and the following covariates from 2015 CPS-3 surveys were identified a priori: age, race/ethnicity (White, Black, Latinx; for sex stratified models), sex (for race/ethnicity stratified models), average daily caloric intake (continuous; estimated from the food frequency questionnaire), comorbidity score (sum of comorbidities including hypertension, hypercholesterolemia, and type 2 diabetes), and height (continuous). Smoking status information was available, however very few participants smoked (2% current smokers), so smoking status was not included in the models.”

Important that authors bring supplementary table 1 and 2 as main tables. Those are important numbers to understand the testing sample.

• Response: We have added the most important information from these supplemental tables, including mean minutes/day of each activity, into the text of the results: “Overall, participants spent 41.6% of the day sedentary (599 min/day), 18.4% in LPA (264 min/day), 4.0% in MPA (57 min/day), 0.6% in VPA (9 min/day), and 35.5% sleeping (511 min/day) on average…”

Regarding the other information presented in Supplemental Tables 2 and 3 (formerly tables 1 and 2), we currently have the maximum number of tables/figures included in the main manuscript and have thus decided to keep the log-ratio variance data as supplemental tables. 

Did authors test for interaction for sex and ethnicity?

• Response: Yes, and we have added a statement to clarify this, “There was significant interaction by race/ethnicity and sex (p = 0.03), though sample sizes were insufficient for joint analyses by sex and race/ethnicity among Latinx and Black participants.”

N for each sub group needs to be defined somewhere clearly, say in your descriptive tables.

• Response: The number of women (n=318) and men (n=231) and Latinx (n=81), Black (n=103), and White (n=365) participants are included in Tables 1 and 2. Each of these tables includes a row for race (which we expanded to include all racial/ethnic groups) and sex, respectively, so that the number of each sex/racial/ethnic group can be calculated: 

Table 1. Baseline characteristics by sex

 Women (n=318) Men (n=231)

Characteristics Loss* (n=111) Maintain (n=57) Gain** (n=150) Loss (n=84) Maintain (n=37) Gain (n=110)

 Arithmetic Mean (SD)

Age (years) 58 (9) 56 (10) 54 (10) 59 (10) 58 (12) 56 (10)

Baseline weight (lbs.) 167 (35) 150 (37) 163 (39) 200 (35) 184 (38) 192 (30)

3-year weight change (lbs.) -9 (12) 0.1 (0.7) 11 (12) -8 (6) 0.1 (0.6) 10 (8)

Energy intake (kcals) 1965 (650) 1914 (741) 1999 (596) 2030 (666) 1856 (631) 1904 (718)

 N (%)

Race/Ethnicity 

 White/Non-Latinx 71 (64%) 41 (72%) 99 (66%) 58 (69%) 28 (76%) 68 (62%)

 Black/Non-Latinx 21 (19%) 9 (16%) 32 (21%) 14 (17%) 4 (11%) 23 (21%)

 Latinx 19 (17%) 7 (12%) 19 (13%) 12 (14%) 5 (13%) 19 (17%)

Current smoker 2 (2%) 2 (4%) 1 (1%) 3 (4%) 1 (3%) 2 (2%)

Baseline body mass index 

 Underweight 0 (0%) 3 (5%) 1 (1%) 0 (0%) 1 (3%) 0 (0%)

 Normal 37 (33%) 35 (61%) 68 (45%) 24 (29%) 15 (41%) 34 (31%)

 Overweight 41 (37%) 13 (23%) 44 (29%) 34 (41%) 16 (43%) 51 (46%)

 Obese 33 (30%) 6 (11%) 37 (25%) 26 (31%) 5 (14%) 25 (23%)

Comorbidity score† 

 0 57 (51%) 38 (67%) 97 (65%) 44 (52%) 14 (38%) 48 (44%)

 1 30 (27%) 12 (21%) 31 (21%) 21 (25%) 13 (35%) 41 (37%)

 2+ 24 (22%) 7 (12%) 22 (15%) 19 (23%) 10 (27%) 21 (19%)

Another way of looking at these numbers has been added as Supplemental Table 1:

 Included (n=549) Excluded (n=201)

Characteristics 

 Arithmetic Mean (SD)

Age (years) 57 (9) 57 (10)

Baseline weight (lbs.) 179 (31) 184 (39)

 N (%)

Women 318 (57.9) 123 (61.1)

White/Non-Latinx 365 (66.6) 167 (83.1)

Current smoker 11 (2.0) 4 (1.9)

Baseline body mass index 

 Underweight 5 (0.9) 2 (1.0)

 Normal 213 (38.8) 75 (37.3)

 Overweight 199 (36.2) 56 (27.9)

 Obese 132 (24.0) 68 (33.8)

Suggest to remove results of joint association. Authors do not have enough sample size to really explore this (for example authors could only do it for whites and not other ethnicities)

• Response: We agree it is unfortunate that our sample size is insufficient for joint analyses among Latinx and Black men and women, we do feel it is important to present the joint analyses we are able to present, as we did indeed see a significant interaction by race/ethnicity and sex (p = 0.03). 

What is rationale for separating vpa keeping in mind it is such a short duration.

• Response: Studies suggest vigorous intensity PA may play an important role in weight loss (references 22 and 23), “To address the potential role of all movement behaviors (especially VPA) in weight loss, [22, 23] a 5-part composition was considered for the main analysis.”

Discussion

Line 216: “Analyses further suggest that time in VPA may be important, relative to other

behaviors, for weight maintenance.” i am not sure which analysis support this result? Your linear reg is your main analysis. That does not explore this association .

• Response: This is in reference to the secondary analyses of relative behaviors by weight change group (presented in the figures). We have clarified that the “secondary analyses” suggest this finding. 

Authors are now using MVPA not MPA and VPA

• Response: We were able to use a 5-part composition for the main analyses, but unfortunately needed to use a 4-part composition for the isotemporal substitution analyses. We added detail to clarify when and why we used 4 vs. 5 part compositions: “For the main analyses, separate multiple linear regression models were first used to examine associations between the 5-part time-use composition (expressed as isometric log-ratios) and three-year weight change (absolute change and percent change modeled continuously) stratified by sex or race/ethnicity and, where feasible, by sex and race/ethnicity. Models were checked to ensure the assumptions of linearity, normality, homoskedasticity and leverage were not violated. A Wald chi square ANOVA type II test of the multiple linear regression models was used to assess the significance of the 5-part time use composition.

Compositional isotemporal substitution analysis was then applied to quantify the change in weight associated with the reallocation of 30 minutes of sedentary time to an equivalent amount of time in another behavior, while keeping time spent in the remaining behaviors constant.[24] Compositional isotemporal substitution analyses explored the replacement of sedentary time with sleep, LPA, or moderate-vigorous physical activity combined (as average time in VPA alone was well under 30 min/day (mean: 13 min/day) and would therefore result in negative time when modeling 30 minutes of replacement).”

Based on the three main principles of compositional analyses, this should not violate any assumptions; according to Dumid, 2017, “The third principle is sub-compositional coherence. The results from a subset of compositional parts must not be reliant on any other parts that are excluded from the subset. In other words, the relative difference between two parts must be the same, regardless of how many parts are included in the analysis.”

---

## [Decision Letter · Decision Letter 1]

14 Dec 2020

Composition of Time in Movement Behaviors and Weight Change in Latinx, Black and White Participants

PONE-D-20-22629R1

Dear Dr. Rees-Punia,

We’re pleased to inform you that your manuscript has been judged scientifically suitable for publication and will be formally accepted for publication once it meets all outstanding technical requirements.

Kind regards,

Sze Yan Liu, PhD

Academic Editor

PLOS ONE

Additional Editor Comments (optional):

Reviewers' comments:

Reviewer's Responses to Questions

**Comments to the Author**

1. If the authors have adequately addressed your comments raised in a previous round of review and you feel that this manuscript is now acceptable for publication, you may indicate that here to bypass the “Comments to the Author” section, enter your conflict of interest statement in the “Confidential to Editor” section, and submit your "Accept" recommendation.

Reviewer #1: All comments have been addressed

2. Is the manuscript technically sound, and do the data support the conclusions?

Reviewer #1: Yes

3. Has the statistical analysis been performed appropriately and rigorously? 

Reviewer #1: Yes

4. Have the authors made all data underlying the findings in their manuscript fully available?

Reviewer #1: No

5. Is the manuscript presented in an intelligible fashion and written in standard English?

Reviewer #1: Yes

6. Review Comments to the Author

Reviewer #1: The study authors have adequately addressed all my comments.

One very minor comment is if instead of p-values if the authors can provide the estimated mean difference and 95% C.I. for the self-reported vs objectively measured weight if possible (line 256)

7. PLOS authors have the option to publish the peer review history of their article (what does this mean?). If published, this will include your full peer review and any attached files.

Reviewer #1: No

---

## [Editor Report · Acceptance letter]

16 Dec 2020

PONE-D-20-22629R1 

Composition of Time in Movement Behaviors and Weight Change in Latinx, Black and White Participants 

Dear Dr. Rees-Punia:

I'm pleased to inform you that your manuscript has been deemed suitable for publication in PLOS ONE. Congratulations! Your manuscript is now with our production department. 

Kind regards, 

on behalf of

Dr. Sze Yan Liu 

Academic Editor

PLOS ONE